# In Vitro Activity of Sulbactam–Durlobactam against Carbapenem-Resistant *Acinetobacter baumannii* Clinical Isolates: A Multicentre Report from Italy

**DOI:** 10.3390/antibiotics11081136

**Published:** 2022-08-22

**Authors:** Bernardetta Segatore, Alessandra Piccirilli, Sabrina Cherubini, Luigi Principe, Giovanni Alloggia, Maria Lina Mezzatesta, Mario Salmeri, Stefano Di Bella, Roberta Migliavacca, Aurora Piazza, Elisa Meroni, Paolo Fazii, Daniela Visaggio, Paolo Visca, Venere Cortazzo, Giulia De Angelis, Arianna Pompilio, Mariagrazia Perilli

**Affiliations:** 1Department of Biotechnological and Applied Clinical Sciences, University of L’Aquila, 67100 L’Aquila, Italy; 2Clinical Pathology and Microbiology Unit, “S. Giovanni di Dio” Hospital, 88900 Crotone, Italy; 3Department of Biomedical and Biotechnological Sciences, Section of Microbiology, University of Catania, 95131 Catania, Italy; 4Clinical Department of Medical, Surgical and Health Sciences, University of Trieste, 34129 Trieste, Italy; 5Unit of Microbiology and Clinical Microbiology, Department of Clinical-Surgical, Diagnostic and Pediatric Sciences, University of Pavia, 27100 Pavia, Italy; 6Clinical Microbiology and Virology Unit, “A. Manzoni” Hospital, 23900 Lecco, Italy; 7Clinical Microbiology and Virology Unit, Spirito Santo Hospital, 65122 Pescara, Italy; 8Department of Science, Roma Tre University, Viale G. Marconi 446, 00146 Rome, Italy; 9Santa Lucia Foundation IRCCS, 00179 Rome, Italy; 10Dipartimento di Scienze Biotecnologiche di Base, Cliniche Intensivologiche e Perioperatorie, Università Cattolica del Sacro Cuore, 00168 Rome, Italy; 11Department of Medical, Oral and Biotechnological Sciences, “G. d’Annunzio” University of Chieti-Pescara, 66100 Chieti, Italy; 12Center of Advanced Studies and Technology (CAST), “G. d’Annunzio” University of Chieti-Pescara, 66100 Chieti, Italy

**Keywords:** durlobactam, *A. baumannii*, WGS

## Abstract

In the present study, the in vitro activity of the sulbactam–durlobactam (SUL–DUR) combination was evaluated against 141 carbapenem-resistant *A. baumannii* (CR*Ab*) clinical strains collected from six Italian laboratories. Over half (54.6%) of these isolates were resistant to colistin. The SUL–DUR combination was active against these CR*Ab* isolates with MIC_50_ and MIC_90_ values of 0.5 mg/L and 4 mg/L, respectively. Only eleven isolates were resistant to SUL–DUR with MIC values ranging from 8 to 128 mg/L. The SUL–DUR resistant *A. baumannii* exhibited several antimicrobial resistance genes (ARGs) such as *bla*_OXA-20_, *bla*_OXA-58_, *bla*_OXA-66_, *bla*_ADC-25_, *aac(6′)-Ib3* and *aac(6′)-Ib-cr* and mutations in *gyr*A (S81L) and *par*C (V104I, D105E). However, in these isolates, mutations Q488K and Y528H were found in PBP3. Different determinants were also identified in these CR*Ab* isolates, including *adeABC*, *adeFGH*, *adeIJK*, *abeS*, *abaQ* and *abaR*, which encode multidrug efflux pumps associated with resistance to multiple antibacterial agents. This is the first report on the antimicrobial activity of SUL–DUR against carbapenem-resistant *A. baumannii* isolates selected from multiple regions in Italy.

## 1. Introduction

*Acinetobacter baumannii* has emerged in the last two decades as one of the major causes of nosocomial infections associated with significant morbidity and mortality and it has been recognized by World Health Organization (WHO) as a “critical priority pathogen” (www.who.int, accesssed on 2 August 2022) [1,2]. *A. baumannii* is ubiquitous and can be found in various environmental sources including soil, water, vegetables, meat and fish [3,4]. In hospital settings, especially in intensive care units, *A. baumannii* can cause ventilator-associated pneumonia and bloodstream infections [5,6,7,8]. The success of this organism is attributed to its ability to survive long-term in hospital environments and its prodigious capacity to acquire new antimicrobial resistance determinants [9]. The mechanisms of resistance in *A. baumannii* include enzymatic inactivation by β-lactamases, modification of target sites (e.g., Penicillin Binding Proteins, PBPs), alterations of porin proteins that result in decreased permeability and the upregulation of the activity of multidrug efflux pumps [9]. Currently, carbapenem-resistant *A. baumannii* (CR*Ab*) pose a global threat to human health. CR*Ab* is emerging worldwide, and the majority of these isolates show multidrug-resistant (MDR), extensively drug-resistant (XDR) and pandrug-resistant (PDR) phenotypes [10,11,12,13]. Currently, few therapeutic options are available for CR*Ab* treatment [14,15]. Generally, colistin (CST), tigecycline and aminoglycosides are used against MDR *A. baumannii*, although there are limitations due to toxicity and poor pharmacokinetic properties [16]. CST has been successfully used to treat pneumoni and, bloodstream and meningitis infections caused by CR*Ab* [17,18]. However, colistin-resistant isolates are emerging worldwide [19]. The intravenous fosfomycin is also used in combination with colistin or tigecycline or aminoglycoside for the treatment of hospital-acquired pneumonia caused by CR*Ab* [20]. Cefiderocol, a novel siderophore cephalosporin, has recently been approved for the treatment of MDR *A. baumannii* [21]. Durlobactam (DUR), previously called ETX2514, is a non-β-lactam diazabicyclooctane (DBO) inhibitor with activity against Ambler class A, C and D β-lactamases [22,23]. Recently, some studies have shown that sulbactam in combination with durlobactam is active against MDR *A. baumannii* [24,25,26,27,28,29,30,31]. Sulbactam (SUL) is one of the first β-lactamase inhibitors used in combination with ampicillin for the treatment of class A β-lactamase-producing pathogens. In *A. baumannii*, SUL also has antibacterial activity by targeting PBPs (i.e., PBP1a/b and PBP3), enzymes required for cell wall synthesis [32]. DUR inactivates serine-β-lactamases by forming a reversible covalent bond with the active site serine [33]. This potent activity of durlobactam allows the susceptibility of CR*Ab* to sulbactam to be restored [22,27]. The aim of the present study was to examine the in vitro activity of sulbactam–durlobactam (SUL–DUR) against 141 CR*Ab* clinical isolates retrospectively collected from six clinical microbiology laboratories located across the national territories representative of northern, central and southern Italy.

## 2. Results

### 2.1. Antimicrobial Susceptibility

Antimicrobial susceptibility of CR*Ab* isolates was previously assessed by each center using commercial systems in the context of normal clinical routine. Consistently, the associated traits were as follows: imipenem and meropenem, MICs > 8 mg/L; gentamicin, MICs > 8 mg/L; ciprofloxacin, MICs > 2 mg/L and SXT, MICs > 8 mg/L (based on trimethoprim concentration). Overall, 64 CR*Ab* isolates were XDR (MIC values for colistin ≤ 2 mg/L), while 77 CR*Ab* isolates showed a PDR phenotype (MIC values for colistin > 2 mg/L).

The in vitro activity of SUL–DUR was evaluated for 141 CR*Ab* clinical isolates using SUL, DUR and CST as comparators. Overall, 77 out 141 (54.6%) *A. baumannii* isolates exhibited resistance to CST with Minimal Inhibitory Concentration (MIC) values of ≥4 mg/L. The MIC_50_ and MIC_90_ for CST were 4 mg/L and >4 mg/L, respectively (Table 1). As shown in Table 1, 131 out of 141 CR*Ab* isolates exhibited MICs > 4 mg/L for SUL and the MIC_50_ and MIC_90_ values were 16 mg/L and 128 mg/L, respectively. DUR had MIC_50_ and MIC_90_ values of 64 mg/L and 128 mg/L, respectively. The SUL–DUR combination was more potent against these CR*Ab* isolates with MIC_50_ and MIC_90_ values of 0.5 mg/L and 4 mg/L, respectively. Only eleven isolates exhibited MIC values > 4 mg/L, the preliminary susceptible breakpoint for SUL–DUR (Table 1) [34,35]. All of the SUL–DUR resistant CR*Ab* isolates were from the Clinical Microbiology Laboratory of Catania University located in southern Italy (Table 2).

### 2.2. Whole-Genome Sequencing of SUL–DUR Resistant A. baumannii: Resistome and Virulome Characterization

Whole-genome sequencing of the eleven SUL–DUR resistant CR*Ab* isolates was performed and these isolates were found to encode several antibiotic resistance genes (ARGs) (Table 3) and virulence-associated genes (VAGs) (Table 4). Among the eleven analyzed strains, all encoded for the class D β-lactamases OXA-20, OXA-58 and OXA-66, in addition to ADC-25, a chromosomally encoded class C β-lactamase. While the presence of these β-lactamases most likely confer resistance to β-lactam antibiotics such as carbapenems, durlobactam has been shown to effectively inhibit these enzymes, so are not a likely cause of the elevated SUL–DUR MICs. Genes that confer resistance to other classes of antibiotics were also detected including aminoglycoside resistance (*aac(6′)-Ib3* and *aac(6′)-Ib-cr*) and fluoroquinolone resistance (mutations in *gyrA* (S81L) and *parC* (V104I, D105E) (Table 2). The Insertion Sequence (IS) IS26 and ISAba125 were also identified in all eleven CR*Ab*, and the transposon Tn6018 was found in two isolates (CT24 and CT58). All of the SUL–DUR-resistant CRAb showed the same profile of virulence factors (Table 4); however, isolates CT57 and CT58 possessed, in addition, *lpsB* and *lpx* VAGs which confer resistance to CST. The eleven CR*Ab* isolates showed the presence of AdeABC, AdeFGH, AdeIJK, abeS, abaQ and abaR multidrug efflux pumps. As shown in Table 4, several genes involved in the biofilm formation system were also identified. Resistance mediated by quorum sensing is represented by *abaI* and *abaR* genes. The *pbp**G*, also known as PBP7, was also identified in all SUL–DUR-resistant CR*Ab* (Table 4). The impact of these genes on SUL–DUR susceptibility is not known.

### 2.3. Molecular Analysis of PBP-3 Gene

The amplification of the PBP-3 gene of the eleven SUL–DUR-resistant *A. baumannii* isolates gave an amplicon of about 1800 bp which was entirely sequenced. In all *A. baumannii* analyzed, the PBP3 showed the following amino acid substitutions: Q488K and Y528H.

## 3. Discussion

The production of carbapenem-hydrolysing β-lactamases is one of the most common mechanisms responsible for carbapenem resistance in *A. baumannii* [10]. Several carbapenemases have been identified in *A. baumannii*, in particular, serine β-lactamases, belonging to class D and metallo-β-lactamases (class B enzymes). Nevertheless, metallo-β-lactamases are very rare in this microorganism [10]. However, in addition to β-lactamases, other mechanisms of resistance to carbapenems, including overexpression of efflux pumps, the reduction or inactivation of the expression of porins and the modification in the expression or synthesis of new PBPs, are found in *A. baumannii* isolates [10]. In the 141 *A. baumannii* analyzed in the present study, class C and D β-lactamases such as OXA-23, OXA-58, OXA-66 (OXA-51-like enzyme), OXA-82 (OXA-51-like enzyme) and ADC-25 were previously identified [36,37,38,39,40,41,42,43,44,45,46,47]. The *bla*_OXA_-types are, usually, flanked by one or two copies of the insertion sequences (i.e., IS*Aba1*, IS*Aba125*) which are located in opposite orientations. These IS*Aba* insertion sequences enhance the expression of *bla*_OXA_ genes and they give genetic plasticity to *A. baumannii* species [48]. Herein, we have demonstrated potent in vitro activity of the SUL–DUR combination against 92% of the CR*Ab* clinical isolates collected from a range of geographical regions within Italy. Of note, SUL–DUR demonstrated antimicrobial activity against both OXA-23- and OXA-58-producing isolates. Moreover, all but two of the colistin-resistant isolates (more than 50% of total isolates) were susceptible to the SUL–DUR combination. Eleven isolates (all from the Microbiology Clinical Laboratory of Catania University) were resistant to SUL–DUR, with most of them belonging to ST2 (*n* = 8) and showing a MIC range of 8–>128 mg/L. Two of these isolates were also resistant to CST. The whole genome analysis of the eleven SUL–DUR-resistant CR*Ab* showed the presence of several ARGs, including *bla*_OXA-20_ (*bla*_OXA-58_ like gene), *bla*_OXA-58_, *bla*_ADC-25_, *aac(6′)Ib-cr*, *aac(6′)-Ib3*, and *tetA*(41), a tetracycline efflux pump protein closely related to *Tet(39)* often found in *Acinetobacter* spp. [49]. Multiple VAGs were also found in these isolates. The multidrug efflux systems (RND, SMR, MFS families) are associated with multiple resistance mechanisms which are capable of extruding a broad range of structurally unrelated compounds [50,51,52]. The contribution of these ARGs and VAGs to the SUL–DUR-resistant phenotype seen in these isolates is not known. However, in the eleven SUL–DUR-resistant *A. baumannii* we found Q488K and Y528H mutations in PBP3. To date, reports of SUL–DUR resistance have been rare and resistance is usually attributed to the presence of metallo-β-lactamases, which DUR does not inhibit, or to mutations near the active site of PBP3, the target of sulbactam [24,25,26]. Few therapeutic choices are available to treat CR*Ab* isolates [14,16,53]. CR*Ab* pneumonia is a major clinical issue with unmet therapeutic needs; in fact, both colistin and tigecycline did not reach a satisfactory epithelial lining fluid concentration and cefiderocol showed disappointing clinical outcomes [54]. DUR displayed an acceptable ratio of epithelial lining fluid to plasma concentrations of 0.37 while SUL reached a 0.5 ratio [34,55]. Another important clinical issue to consider is that the SUL–DUR combination is expected to have a lower degree of nephrotoxicity compared to CST (ATTACK Trial. Available online: https://investors.entasistx.com/news-releases/news-release-details/entasis-therapeutics-announces-positive-topline-results, accessed on 2 August 2022), and in general be more safe. Taking into account the significant MIC reduction reached upon adding DUR to SUL and the promising clinical data from the phase 3 clinical trial comparing the safety and efficacy of SUL–DUR to CST for the treatment of infections caused by CR*A**b* (SUL–DUR mortality 19%, CST mortality 32%, 95% CI: −30.0, 3.5), if approved, SUL–DUR may be an important option for CR*Ab* treatment regimens. Further studies are needed to elucidate the molecular mechanisms responsible for resistance to SUL-DUR and to explore its therapeutic potential. It will also be necessary to combine in vitro findings with additional pharmacokinetic and pharmacodynamic data in order to provide more meaningful predictions of the in vivo efficacy of SUL-DUR combination in clinical practice.

## 4. Materials and Methods

### 4.1. Antibiotics and Inhibitors

SUL and DUR (ETX2514) were kindly provided by Dr. Alita A. Miller, Entasis Therapeutics, Waltham, MA, USA.

### 4.2. Bacterial Strains Selection

A total of 141 non duplicate CR*Ab* strains, which were previously characterized for their mechanisms of resistance, were included in this study [36,37,38,39,40,41,42,43,44,45,46,47]. In particular, we retrospectively selected 64 XDR and 77 PDR previously characterized CR*Ab* clinical isolates collected in six centers from Italy. Isolates were selected on the basis of this extensive resistance although resistance determinants were different (mostly including OXA-type enzymes, as reported in related publications). Most of the isolates (121 out of 141) were collected from five clinical microbiology laboratories distributed throughout northern, central and southern Italy. Specifically, 16 *A. baumannii* were collected from the Microbiology Laboratory of the University of Pavia (Northern Italy) during 2018, 26 isolates were collected from the teaching “Gemelli” Hospital Rome (central Italy) between 2020 and 2022, 8 isolates were collected from Spirito Santo Hospital Pescara (Central Italy) in 2020–2021, 1 isolate was from the University of L’Aquila (Central Italy) and 70 isolates were collected from the Clinical Microbiology Laboratory of the University of Catania (Southern Italy) between 2008 and 2018. In addition, 20 *A. baumannii* were kindly given by Professor Visca, Clinical Microbiology Laboratory of Roma Tre University (Central Italy). These strains were collected in 2004–2014 from different countries during the project “Carbapenem-resistant *Acinetobacter baumannii*: whole-genomic and phenomic investigation of the traits that favored the predominance and shift to OXA-23-producing IC2 isolates”; funded by ESCMID in 2017. The antimicrobial susceptibility of CR*Ab* isolates was previously assessed by participating centers using commercial systems in the context of normal clinical routine. Tested antibiotics were imipenem, meropenem, gentamicin, ciprofloxacin and colistin. All strains were collected from different wards including intensive care units, infectious diseases units, neurosurgery, pneumology, thoracic surgery and internal medicine. All *A. baumannii* were isolated from clinical sources, including sputum, blood, urine, wounds, peritoneal fluid, liquor and stool. The *A. baumannii* isolates belonged to the following sequence types: ST2 (*n* = 121), ST1 (*n* = 6), ST4 (*n* = 1), ST20 (*n* = 5), ST78 (*n* = 2), ST81 (*n* = 1), ST95 (*n* = 1), ST109 (*n* = 1), ST196 (*n* = 1) and ST197 (*n* = 1). In these *A. baumannii* strains, the carbapenem resistance was mainly mediated by the presence of OXA-23 (80 isolates), OXA-58 (48 isolates), OXA-66 (4 isolates) and OXA-82 (4 isolates). The simultaneous presence of OXA-23 and OXA-58 was found in 11 isolates and, in addition, ADC-25, a chromosomal AmpC enzyme, was also identified [36,37,38,39,40,41,42,43,44,45,46,47].

MDR isolates were defined as those with acquired non-susceptibility to at least one agent in three or more antimicrobial categories. XDR isolates were defined as those with acquired non-susceptibility to all antibiotics, except for one or two. PDR isolates were defined as those with acquired non-susceptibility to all antibiotics.

### 4.3. Bacterial Strains Identification

*A. baumannii* isolates were collected by standard methods, followed by isolation in pure culture on MacConkey agar plates, identified by the Vitek 2 system (bioMerieux, Marcy l’Etoile, France) and stored in Brain Hearth Infusion broth with 15% glycerol and frozen at −80 °C.

### 4.4. MIC Determination

The MIC experiments were performed by conventional broth microdilution procedures in Mueller Hinton broth supplemented with calcium and magnesium to physiological concentrations (CAMHB), using an inoculum of 5 × 10^5^ CFU/mL according to the Clinical and Laboratory Standards Institute (CLSI) [56]. One hundred and forty-one non-duplicate *A. baumannii* isolates were tested against DUR alone as well as SUL alone, plus SUL–DUR and CST. For SUL, a susceptibility breakpoint of 4 mg/L was used, based on the CLSI ampicillin–sulbactam susceptible breakpoint of 8/4 mg/L for *Acinetobacter spp* [56]. SUL–DUR MICs were performed as 2-fold dilutions of SUL with DUR at a fixed concentration of 4 mg/L [56]. MICs were interpreted using CLSI breakpoints where available. Concurrent quality control (QC) procedures were performed by testing *Escherichia coli* ATCC 25922, examined for each MIC run. Following 18 to 20 h of aerobic incubation at 37 °C, the microplates were examined for growth. The determination of all MICs was performed in three separate sets of experiments.

### 4.5. Whole-Genome Sequencing

Total nucleic acid was extracted using MagMax Microbiome Ultra Nucleic Acid Isolation kit (Applied Biosystems and ThermoFisher Scientific, Monza, Italy). Genomic libraries were prepared using Swift 2S Turbo DNA Library kit (Swift Biosciences, Ann Arbor, MI, USA) as previously reported [57,58]. WGS was performed on an Illumina MiSeq using v3 reagent kits generating 2 × 300 bp paired-end reads (Illumina, San Diego, CA, USA). DRAGEN FastQC + MultiQC v3.9.5 (https://basespace.illumina.com/apps/10562553/DRAGEN-FastQC-MultiQC, accessed on 24 May 2022) were used for quality control and sequences filtering. Paired-end reads were assembled with Velvet de novo Assembly v1.0.0 (https://basespace.illumina.com/apps/8556549/Velvet-de-novo-Assembly, accessed on 5 June 2022). Multi-Locus Sequence Typing (MLST) on assembled *A. baumannii* genomes was performed according to the Pasteur scheme. ResFinder4.1 and MobileElementFinder 1.0.3 were used to detect acquired antimicrobial resistance genes and mobile genetic elements, respectively. ResFinder and MobileElementFinder 1.0.3 databases were synchronized with databases from Center for Genomic Epidemiology (http://www.genomicepidemiology.org/, accessed on 10 June 2022). Virulence Factor Database (VFDB) was used for the detection of virulence genes (http://www.mgc.ac.cn/VFs/, accessed on 2 August 2022).

### 4.6. PBP-3 Amplification and Sequencing

The amplification of the PBP-3 gene was performed in PCR using the total genome of the SUL–DUR-resistant *A. baumannii* (CT20, CT24, CT25, CT26, CT29, CT30, CT31, CT32, CT57, CT58, CT68) and the following external primers: PBP-3_F 5′TTACCTGCGAATAGGATTTTCTG and PBP-3_R 5′ ATGTGGCGGTTTTATCTGCT. The amplicons obtained were purified and directly sequenced on both strands by using a BigDye Sequencing Reaction Kit and an ABI PRISM 3500 capillary automated sequencer (Applied Biosystem, Monza, Italy).

## 5. Conclusions

In the present study, SUL–DUR demonstrated good in vitro antimicrobial activity against XDR and PDR *A. baumanni* clinical isolates, collected from different regions across Italy. These data confirmed the results from recent studies showing good activity of the SUL–DUR combination against MDR, XDR and PDR *A. baumannii* [59]. To the best our knowledge, this study also represents the first report on SUL–DUR activity against a large number of carbapenem-resistant, and largely colistin-resistant, *A. baumannii* isolates from Italy.

## Figures and Tables

**Table 1 antibiotics-11-01136-t001:** In vitro activities of sulbactam–durlobactam and comparators against 141 carbapenem-resistant *Acinetobacter baumannii* collected in Italy.

Antimicrobial Agent	Number of Isolates at Each MIC (mg/L)																
	0.06	0.125	0,25	0.5	1	2	4	>4	8	16	32	64	128	>128	MIC RANGE	MIC_50_	MIC_90_
SUL	/	/	/	/	/	2	8	_	27	45	33	8	13	5	0.06–>128	16	128
DUR	/	/	/	/	/	/	/	_	3	7	44	39	47	1	0.06–>128	64	128
SUL–DUR	/	4	25	51	30	14	6	_	4	2	/	/	/	5	0.06/4–>128/4	0.5	4
CST	/	7	6	12	20	19	22	55	_	_	_	_	_	_	0.06–>4	4	>4

/ = the number of isolates equal to zero. _, no values available. In SUL–DUR combination, DUR was at fixed concentration of 4 mg/L.

**Table 2 antibiotics-11-01136-t002:** MIC distribution of sulbactam–durlobactam and comparators against 141 CR*Ab* isolates by location of the clinical microbiology laboratory.

City (No. Isolates) Antimicrobial Agents	Number of Isolates with MIC (mg/L)													
	0.06	0.125	0.25	0.5	1	2	4	>4	8	16	32	64	128	>128
**Pavia (16)**														
SUL	/	/	/	/	/	/	1	-	1	5	7	2	/	/
DUR	/	/	/	/	/	/	/	-	/	/	/	2	13	1
SUL–DUR	/	/	/	6	6	4	/	-	/	/	/	/	/	/
CST	/	/	1	6	9	/	/	/	-	-	-	-	-	-
**Gemelli (26)**														
SUL	/	/	/	/	/	1	2	-	4	11	7	1	/	/
DUR	/	/	/	/	/	/	/	-	/	2	6	11	7	/
SUL–DUR	/	/	3	16	7	/	/	-	/	/	/	/	/	/
CST	/	/	/	/	/	/	13	13	-	-	-	-	-	-
**PE/AQ (9)**														
SUL	/	/	/	/	/	/	/	-	/	1	6	1	1	/
DUR	/	/	/	/	/	/	/	-	/	/	3	2	4	/
SUL–DUR	/	/	1	2	2	4	/	-	/	/	/	/	/	/
CST	/	/	/	/	/	/	/	9	-	-	-	-	-	-
**Roma Tre (20)**														
SUL	/	/	/	/	/	1	3	-	3	5	7	1	/	/
DUR	/	/	/	/	/	/	/	-	/	/	6	6	8	/
SUL–DUR	/	/	3	7	7	1	2	-	/	/	/	/	/	/
CST	/	/	/	/	/	7	7	6	-	-	-	-	-	-
**Catania (70)**														
SUL	/	/	/	/	/	/	2	-	19	23	6	3	12	5
DUR	/	/	/	/	/	/	/	-	3	5	29	18	15	/
SUL–DUR	/	4	18	20	8	5	4	-	4	2	/	/	/	5
CST	/	7	5	6	11	12	2	27	-	-	-	-	-	-

**Pavia**, isolates collected from the Microbiology Laboratory of the University of Pavia. **Gemelli**, isolates collected from the teaching “Gemelli” Hospital Rome. **PE/AQ**, isolates collected from Spirito Santo Hospital Pescara and the University of L’Aquila. **Roma Tre**, isolates collected from the Clinical Microbiology Laboratory of Roma Tre University. **Catania**, isolates collected from the Clinical Microbiology Laboratory of the University of Catania./= the number of isolates equal to zero. _, no values available. In the SUL–DUR combination, DUR was at a fixed concentration of 4 mg/L.

**Table 3 antibiotics-11-01136-t003:** Characterization of Sulbactam–Durlobactam-resistant *A. baumannii*.

Strain	Sequence Type	Ward	Sample	SUL–DUR MIC (mg/L)	SUL MIC (mg/L)	DUR MIC (mg/L)	CST MIC (mg/L)	Resistance Genes		Mobile Genetic Elements
								β-Lactamases	Other	
*A. baumannii* CT20	2	transplant	BAL	8	128	128	0.125	*bla* _ADC-25_ *bla* _OXA-20_ *bla* _OXA-58_	*aac(6′)-Ib-cr* *aac(6′)-Ib3* *tetA(41)*	IS*26*, IS*Aba125*
*A. baumannii* CT57	2	ICU	BAL	8	128	64	64	bla_ADC-25_ bla_OXA-20_ bla_OXA-58_	*aac(6′)-Ib-cr* *aac(6′)-Ib3* *tetA(41)*	Tn*6018*, IS*26*, IS*Aba125*
*A. baumannii* CT58	2	ICU	wound	8	128	32	32	bla_ADC-25_ bla_OXA-20_ bla_OXA-58_	*aac(6′-)Ib-cr* *aac(6′)-Ib3* *tetA(41)*	Tn*6018*, IS*26*, IS*Aba125*
*A. baumannii* CT68	2	ICU	blood	8	128	64	0.25	bla_ADC-25_ bla_OXA-20_ bla_OXA-58_	*aac(6′)-Ib-cr* *aac(6′)-Ib3* *sul1*	IS*26*, IS*Aba125*
*A. baumannii* CT24	2	ICU	blood	16	64	128	0.5	bla_ADC-25_ bla_OXA-20_ bla_OXA-58_ *bla* _OXA-66_	*aac(6′)-Ib-cr* *aac(6′)-Ib3* *qacE* *sul1* *gyrA* (S81L) *parC*(V104I, D105E)	IS*26*, IS*Aba125*
*A. baumannii* CT25	2	ICU	catheter	16	128	64	1	bla_ADC-25_ bla_OXA-20_ bla_OXA-58_ bla_OXA-66_	*aac(6′)-Ib-cr* *aac(6′)-Ib3* *sul1*	IS*26*, IS*Aba125*
*A. baumannii* CT26	2	surgery	bile	>128	>128	64	0.5	bla_ADC-25_ bla_OXA-20_ bla_OXA-58_ bla_OXA-66_	*aac(6′)-Ib-cr* *aac(6′)-Ib3* *qacE* *sul1* *gyrA* (S81L) *parC* (V104I, D105E)	IS*26*, IS*Aba125*
*A. baumannii* CT29	2	ICU	exudate	>128	>128	128	1	bla_ADC-25_ bla_OXA-20_ bla_OXA-58_ bla_OXA-66_	*aac(6′)-Ib-cr* *aac(6′)-Ib3* *qacE* *sul1* *gyrA* (S81L) *parC* (V104I, D105E)	IS*26*, IS*Aba125*
*A. baumannii* CT30	20	ICU	catheter	>128	>128	32	1	bla_ADC-25_ bla_OXA-20_ bla_OXA-58_ bla_OXA-66_	*aac(6′)-Ib-cr* *aac(6′)-Ib3* *qacE* *sul1* *gyrA* (S81L) *parC* (V104I, D105E)	IS*26*, IS*Aba125*
*A. baumannii* CT31	20	ICU	pus	>128	>128	128	0.125	bla_ADC-25_ bla_OXA-20_ bla_OXA-58_ bla_OXA-66_	*aac(6′)-Ib-cr* *aac(6′)-Ib3* *qacE* *sul1* *gyrA* (S81L) *parC*(V104I, D105E)	IS*26*, IS*Aba125*
*A. baumannii* CT32	20	ICU	BAL	>128	>128	128	1	bla_ADC-25_ bla_OXA-20_ bla_OXA-58_ bla_OXA-66_	*aac(6′)-Ib-cr* *aac(6′)-Ib3* *qacE* *sul1* *gyrA* (S81L) *parC* (V104I, D105E)	IS*26*, IS*Aba125*

**Table 4 antibiotics-11-01136-t004:** Virulence factors encoded by the eleven *A. baumannii* isolates resistant to SUL–DUR.

SUL–DUR-Resistant *A. baumannii* (Strains No.: CT20, CT24, CT25, CT26, CT29, CT30, CT31, CT32, CT57, CT58, CT68)	
Virulence-Associated Genes	Virulence Factors
*adeA, adeC, adeF, adeG, adeH, adeI, adeK, adeL, adeN, adeJ, adeR*	RND efflux pump *AdeABC*, *AdeFGH* and *AdeIJK*
*abeS*	SMR family of transporter efflux pumps
*abaQ, abaF*	MFS transporters
*plc, plcD*	Phospholipase
*lpsB* (only in CT57 and CT58)	Lipopolysaccharide synthesis (mutations are involved in CST resistance)
*lpxA, lpxB, lpxC, lpxD, lpxL, lpxM* (only in CT57 and CT58)	Biosynthesis of lipid A (mutations are involved in CST resistance)
*barA, barB* *basA, basB, basC, basD, basF, basG, basH, basI, basJ* *bauA, bauB, bauC, bauD, bauE, bauF* *entE* *hemO*	Iron uptake: acinetobactin and heme utilization
*bap, pgaA, pgaB, pgaC, pgaD, csuA, csuB, csuC, csuD, csuE, bfmR, bfmS*	Biofilm formation system and cell–cell adhesion
*abaI, abaR*	Quorum sensing
*pbpG (or PBP7) and PBP3^Q488K^ and PBP3^Y528H^*	Penicillin-binding protein
*katA*	A secondary catalase/peroxidase

RND, resistant-nodulation division. SMR, small multidrug resistance. MFS, major facilitator superfamily.

## Data Availability

Not applicable.

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
