# Peer review of "In Vitro Activity of Sulbactam–Durlobactam against Carbapenem-Resistant Acinetobacter baumannii Clinical Isolates: A Multicentre Report from Italy"

_antibiotics, 2022, doi:10.3390/antibiotics11081136_

Round 1

Reviewer 1 Report

Multi-drug resistant bacteria represent a challenge for the physician and finding genetic mutations that induce antibiotic resistance can be used in the future for surveillance. Several articles on the same subjects have been published already and are also found between references.

Proving that Sulbactam-Durlobactam has a beneficial effect against carbapenem resistant Acinetobacter baumannii clinical isolates confirms former similar studies. 

This article is well written, interesting and results are clearly presented. Discussion section is consistent and references are well chosen.

Author Response

Thank you for the nice comment

Reviewer 2 Report

In this article, the authors present a multicentre study characterizing the in vitro activity of sulbactam-durlobactam against A.baumannii. Overall, the manuscript is scientifically sound and very well-written, and I dont have more comments or suggestions to modify this manuscript from its present form. 

Author Response

Thank you for the nice comment

Reviewer 3 Report

This manuscript fitted well in the scope of journals. The timely update of AMR in bacterial pathogen is necessary and important. Authors nicely addressed the issue of CRAB in the manuscript with the solution of sul-dur combination therapy against the CRAB infection. However, there are some limitations and major correction are required in the manuscript before its publication. 

In my opinion manuscript might be accepted for publication provided that some additional data and minor correction. 

Major comments

1. It would be great to have a data on presence or absence of antibiotic resistance gene that confer the resistance to that respective class or antibiotic. The panel of antibiotics is missing, authors demonstrated only MIC of sul, dur, and cst. Showing presence/absence of resistance gene without the MIC for that antibiotic makes no sense. Authors should take atleast one or two antibiotic from the each class  and correlate the data with the antibiotic resistance gene that you already included here.   

2. The experiment lacks control ATCC strains in experiment.

3. Authors should do in vitro synergy study that will confirm both the drugs synergizes without antagonism. Please do chequerboard assay (FIC data) for sul-dur (Reference FC Odd paper doi: 10.1093/jac/dkg301. ).  Sul-Dur combination already shown the resistance in 11 strains of CRAB, this combination may not worked in animal. To move forward with this combination, we must need a chequerboard assay FIC data. How will authors project this combination further, please comment. 

4. Conclusion does not support the results, 'Sul-dur have potent activity against MDR, XDR and PDR strains' its not correct.

Minor Comments

1. Line 54: Please change 'critical' to "critical priority pathogen".

2.Line 65: Please define MDR, XDR and PDR once in the MS.

Author Response

This manuscript fitted well in the scope of journals. The timely update of AMR in bacterial pathogen is necessary and important. Authors nicely addressed the issue of CRAB in the manuscript with the solution of sul-dur combination therapy against the CRAB infection. However, there are some limitations and major correction are required in the manuscript before its publication.

In my opinion manuscript might be accepted for publication provided that some additional data and minor correction.

Major comments

  1. It would be great to have a data on presence or absence of antibiotic resistance gene that confer the resistance to that respective class or antibiotic. The panel of antibiotics is missing, authors demonstrated only MIC of sul, dur, and cst. Showing presence/absence of resistance gene without the MIC for that antibiotic makes no sense. Authors should take at least one or two antibiotic from the each class and correlate the data with the antibiotic resistance gene that you already included here.

R: The aim of our study was to assess the antimicrobial activity of SUL/DUR combination against XDR and PDR Carbapenem-Resistant A. baumannii (CRAB) clinical isolates. Thus, we retrospectively selected 64 XDR and 77 PDR previously characterized CRAB clinical isolates collected in six centers from Italy. Isolates were selected on the basis of this extensive resistance although resistance determinants were different (mostly including OXA-type enzymes, as reported in related publications). Concerning antimicrobial susceptibility, it was previously assessed by each center using commercial systems in the context of normal clinical routine. Consistently, associated traits were as follows: imipenem and meropenem, MICs >8 mg/L; gentamicin, MICs > 8 mg/L; ciprofloxacin, MICs >2 mg/L and SXT, MICs >8 mg/L (based on trimethoprim concentration). This point has now been clarified in M&M and Results. On the other hand, the antimicrobial activity of sulbactam, durlobactam, colistin and sulbactam/durlobactam combination has been assessed in this study by broth microdilution reference methods.

Whole genome sequencing has been performed only for SUL/DUR-resistant isolates. Also in this case, the antimicrobial susceptibility has been previously assessed by participant centers, showing nine XDR and two PDR A. baumannii isolates.

  1. The experiment lacks control ATCC strains in experiment.

R: Escherichia coli ATCC 25922 has been used as quality control strain in MIC determination assays, as mentioned in M&M, paragraph 4.4.

  1. Authors should do in vitro synergy study that will confirm both the drugs synergizes without antagonism. Please do chequerboard assay (FIC data) for sul-dur (Reference FC Odd paper doi: 10.1093/jac/dkg301. ). Sul-Dur combination already shown the resistance in 11 strains of CRAB, this combination may not worked in animal. To move forward with this combination, we must need a chequerboard assay FIC data. How will authors project this combination further, please comment.

R: The susceptibility testing of SUL with DUR, at a fixed concentration of 4 mg/L, was performed as recommended by the manufacturer (Entasis Therapeutics, Waltham, Massachusetts, USA).  

The SUL-DUR is an investigational drug, being developed as monotherapy for the treatment of infections caused by Acinetobacter baumannii, including carbapenem-resistant strains. It is a combination of sulbactam, an IV β-lactam antibiotic, and durlobactam, a novel broad-spectrum IV β-lactamase inhibitor added to sulbactam at a fixed concentration of 4 μg/ml (Barnes MD, Kumar V, Bethel CR, Moussa SH et al. 2019. Targeting multidrug-resistant Acinetobacter spp.: sulbactam and the diazabicyclooctenone β-lactamase inhibitor ETX2514 as a novel therapeutic agent. mBio 10:e00159-19. https:// doi.org/10.1128/mBio.00159-19).

Here we report published studies, using checkerboard assays, that were performed to test Sulbactam/Durlobactam in combination with other antibiotics:

  • Petropoulou D, Siopi M, Vourli S and Pournaras S (2022) Activity of Sulbactam- Durlobactam and Comparators Against a National Collection of Carbapenem-Resistant Acinetobacter baumannii Isolates From Greece. Front. Cell. Infect. Microbiol. 11:814530. doi: 10.3389/fcimb.2021.814530
  • Dana J. Holger,1 Ashlan J. Kunz Coyne,1 Jing J. Zhao,2 Avnish Sandhu,3,4 Hossein Salimnia,5 and Michael J. Rybak1,4, Novel Combination Therapy for Extensively Drug-Resistant Acinetobacter baumannii Necrotizing Pneumonia Complicated by Empyema: A Case Report Open Forum Infect Dis. 2022 Mar 5;9(4):ofac092.  doi: 10.1093/ofid/ofac092. eCollection 2022 Apr.
  • Zaidan N, Hornak JP, Reynoso D. 2021. Extensively drug-resistant Acinetobacter baumannii nosocomial pneumonia successfully treated with a novel antibiotic combination. Antimicrob Agents Chemother 65:e00924-21. https://doi.org/10.1128/AAC.00924-21.
  • Carter, S. McLeod, A. Shapiro, and A. Miller. In Vitro Activity of Sulbactam-Durlobactam in Combination with Other Antimicrobial Agents. Poster 2776 Presented at ASM Microbe 2022, Washington, DC

While these early results are encouraging, additional studies could be carried out to evaluate the in vitro activity of SUL/DUR in combination with other antibiotics against our SUL/DUR-resistant A.baumnnii.

Regarding “this combination may not worked in animal”: it is well known that in vitro effectiveness does not imply in vivo efficacy (animal, humans), our study is just an in vitro one, other scientists will assess clinical efficacy. However, SUL/DUR combination is currently in Phase III trial for Acientobacter infections, so hopefully we will not wait long for clinical results. A comment on the need of further studies to better understand the potential therapeutic role of SUL/DUR combination has been added in the Discussion section.

  1. Conclusion does not support the results, 'Sul-dur have potent activity against MDR, XDR and PDR strains' is not correct.

R: We have added now these sentences:

M&M – “In particular, we retrospectively selected 64 XDR and 77 PDR previously characterized CRAB clinical isolates collected in five six centers from Italy. Isolates were selected on the basis of this extensive resistance although resistance determinants were different (mostly including OXA-type enzymes, as reported in related publications)”.

M&M – “The antimicrobial susceptibility of CRAB isolates was previously assessed by participating centers using commercial systems in the context of normal clinical routine. Tested antibiotics were imipenem, meropenem, gentamicin, ciprofloxacin and colistin”.

Results – “Antimicrobial susceptibility of CRAB isolates was previously assessed by each center using commercial systems in the context of normal clinical routine. Consistently, associated traits were as follows: imipenem and meropenem, (MICs >8 mg/L); gentamicin, (MICs > 8 mg/L); ciprofloxacin, (MICs >2 mg/L) and SXT, (MICs >8 mg/L, based on trimethoprim concentration). Overall, 64 CRAB isolates were XDR (MIC values for colistin ≤2), while 77 CRAB isolates showed a PDR phenotype (MIC values for colistin >2)”.

Furthermore, we have deleted “potent”. In our study SUL/DUR combination showed good antimicrobial activity against 55 XDR and 75 PDR A. baumannii clinical isolates.

Minor Comments

  1. Line 54: Please change 'critical' to "critical priority pathogen".

R: Done.

2.Line 65: Please define MDR, XDR and PDR once in the MS.

R: Done (in M&M).

Round 2

Reviewer 3 Report

No comments.